# Microgravity Affects Thyroid Cancer Cells during the TEXUS-53 Mission Stronger than Hypergravity

**DOI:** 10.3390/ijms19124001

**Published:** 2018-12-12

**Authors:** Sascha Kopp, Marcus Krüger, Johann Bauer, Markus Wehland, Thomas J. Corydon, Jayashree Sahana, Mohamed Zakaria Nassef, Daniela Melnik, Thomas J. Bauer, Herbert Schulz, Andreas Schütte, Burkhard Schmitz, Hergen Oltmann, Stefan Feldmann, Manfred Infanger, Daniela Grimm

**Affiliations:** 1Clinic for Plastic, Aesthetic and Hand Surgery, Otto von Guericke University Magdeburg, Leipziger Str. 44, D-39120 Magdeburg, Germany; sascha.kopp@med.ovgu.de (S.K.); marcus.krueger@med.ovgu.de (M.K.); markus.wehland@med.ovgu.de (M.W.); mohamed.nassef@med.ovgu.de (M.Z.N.); daniela.melnik@med.ovgu.de (D.M.); thomas.bauer@med.ovgu.de (T.J.B.); manfred.infanger@med.ovgu.de (M.I.); 2Max Planck Institute of Biochemistry, D-82152 Martinsried, Germany; jbauer@biochem.mpg.de; 3Department of Biomedicine, Aarhus University, Wilhelm Meyers Allé 4, DK-8000 Aarhus C, Denmark; corydon@biomed.au.dk (T.J.C.); jaysaha@biomed.au.dk (J.S.); 4Department of Ophthalmology, Aarhus University Hospital, Aarhus, 8000 Aarhus C, Denmark; 5Cologne Center for Genomics, University of Cologne, D-50931 Cologne, Germany; herbert.schulz@uni-koeln.de; 6Airbus Defence and Space GmbH, Airbus-Allee 1, D-28199 Bremen, Germany; andreas.schuette@airbus.com (A.S.); burkhard.schmitz@airbus.com (B.S.); hergen.oltmann@airbus.com (H.O.); stefan.feldmann@airbus.com (S.F.); 7Gravitational Biology and Translational Regenerative Medicine, Faculty of Medicine and Mechanical Engineering, Otto-von-Guericke-University Magdeburg, D-39120 Magdeburg, Germany

**Keywords:** Microgravity, hypergravity, random positioning machine, thyroid cancer, sounding rocket, cytoskeleton, focal adhesion, extracellular matrix

## Abstract

Thyroid cancer is the most abundant tumor of the endocrine organs. Poorly differentiated thyroid cancer is still difficult to treat. Human cells exposed to long-term real (r-) and simulated (s-) microgravity (µ*g*) revealed morphological alterations and changes in the expression profile of genes involved in several biological processes. The objective of this study was to examine the effects of short-term µ*g* on poorly differentiated follicular thyroid cancer cells (FTC-133 cell line) resulting from 6 min of exposure to µ*g* on a sounding rocket flight. As sounding rocket flights consist of several flight phases with different acceleration forces, rigorous control experiments are mandatory. Hypergravity (hyper-*g*) experiments were performed at 18*g* on a centrifuge in simulation of the rocket launch and s-µ*g* was simulated by a random positioning machine (RPM). qPCR analyses of selected genes revealed no remarkable expression changes in controls as well as in hyper-*g* samples taken at the end of the first minute of launch. Using a centrifuge initiating 18*g* for 1 min, however, presented moderate gene expression changes, which were significant for *COL1A1*, *VCL*, *CFL1*, *PTK2*, *IL6,*
*CXCL8* and *MMP14*. We also identified a network of mutual interactions of the investigated genes and proteins by employing *in-silico* analyses. Lastly, µ*g*-samples indicated that microgravity is a stronger regulator of gene expression than hyper-*g*.

## 1. Introduction

Thyroid cancer (TC) is one of the most abundant neoplasms of the endocrine organs [1]. Thyroid carcinomas are malignant tumors of the thyroid gland and are classified into differentiated TC (papillary TC, follicular TC and Hürthle cell cancer), medullary TC; and anaplastic TC. The American Cancer Society recently estimated that in the US in 2018 about 2,060 persons died from TC [1].

Thyroid cells as well as differentiated TC cells are capable of taking up iodine [2]. When radioactive iodine (RAI; ^131^I) is given, it concentrates in the thyroid. The radiation will destroy the thyroid gland and any other benign and malignant thyroid cells. RAI-treatment is used to ablate any thyroid tissue in the organism. Unfortunately, undifferentiated TC cells lost the ability to take up iodine which increases the need to find new therapeutically valuable treatments [3], for example by targeting tumor angiogenesis [2,4].

Microgravity (µ*g*) offers a very special culture environment for cells enabling three-dimensional (3D) growth. This environment is suitable for tissue engineering purposes, while exposing cells to an unknown stress and thereby forcing them to react and adapt to this new condition [5,6,7].

Normal and benign thyroid cells have been examined after cultivation on a random positioning machine (RPM) at various time-points [8,9,10]. The results suggest that RPM-exposure changed differentiation and growth behavior of poorly differentiated malignant FTC-133 cells after long-term RPM-exposure [8]. Comparable results were obtained from the Simbox/Shenzhou-8 mission, when FTC-133 cells were cultured on an unmanned spacecraft for 10 days [11].

Furthermore, the cytoskeleton underwent changes in structure and organization by exposure to µ*g*. This was demonstrated during parabolic flights as well as a TEXUS (TX) sounding rocket mission, where a novel live-cell imaging approach using the FLUMIAS microscope revealed the occurrence of “holes” in the F-actin-network and in lamellipodia- and filopodia-like structures [12]. According to the tensegrity model, meaning that cells gain their structure through a pervasive tensional force, the cytoskeleton might play an important role in the mechano- and gravisensing process of human cells [13,14]. It is proposed that the cytoskeleton acts as a mediator and conductor of signals from the extracellular space into the nucleus due to it being hardwired to specific cell surface proteins, such as integrins, which are involved in focal adhesion formation and cell-cell interaction [15,16], as well as to cytoplasmic transduction molecules [13]. Proteins involved in the linkage of the cytoskeleton to membrane proteins and the ECM are most prominently talin, vinculin or the ERM proteins [17,18,19].

To determine the impact of r-µ*g* on the cells after the 1-min long hyper-*g* phase during launch-, FTC-133 cells were exposed to 6 min of r-µ*g* during the TEXUS-(TX) 53 mission [20] and in parallel to 1 min of 18*g* on a centrifuge (see Figure 1 of [20]). In addition, we compared the effects of the short-term r-µ*g* with a 6-min cultivation on the RPM (s-µ*g*), focusing on extracellular matrix (ECM) proteins, the cytoskeleton, cell adhesion, focal adhesion and cytokines.

The expression of genes and the accumulation of proteins, which were known to be highly sensitive to altered gravity conditions were analyzed. Furthermore, we have conducted an *in-silico* analysis including data from the literature to identify possible mutual interaction networks between the analyzed proteins, as successfully practiced earlier [21,22].

## 2. Results

After the samples’ return to the laboratory, quantitative real-time PCR (qPCR) and pathway analyses were performed. The studies were done on the samples collected during the TX53 mission as described in [20] as well as on samples exposed to 18*g* (worst case) hyper-g on a centrifuge, or cultured for 6-min of s-µ*g* exposure on an RPM. We focused on the F-actin cytoskeleton, the ECM, focal adhesion, cell adhesion molecules, and cytokines. Various genes belonging to these groups and recently found to be differentially expressed during long-term spaceflights and RPM-experiments (Table 1) were investigated.

The genes listed in Table 1 were complemented with a number of additional genes known to be involved in cell-cell and cell-substrate adhesion. The *in-silico* evaluation of the selected genes revealed a self-contained network. These genes, indicated by a green or blue rim, form a network of mutual influence (Figure 1, icons with red rim), with *TGFB1*, *ITGB1*, *SPP1*, *IL-6*, *IL8*, *FN1*, *COL1A1*, *CDH1*, *and MMP14* being nodal points of this network. Additionally, many arrows start at *TGFB1*, *LOX* and *PTK2*, meaning that these entities are effectors in the system. Interestingly, the 6 genes marked with a yellow rim fit rather well into the network. These genes caught our attention during a preliminary evaluation of the microarray analysis of the two µ*g*-samples available, because they were at least 1.7-fold enhanced in comparison to the controls. 

Regarding the products of these genes (Figure 2), three groups can be distinguished: There are extracellular proteins, membrane proteins and intracellular proteins. As indicated above, the proteins whose icons are marked with a green or blue rim, are gene products found in recent studies referenced or mentioned in the preceding paper [20], respectively. On the protein level, however, the most important nodal points are fibronectin and integrin β 1. Intracellularly, they are connected to VCL, MSN and ACTB, which are involved in forwarding signals, generated by cells’ binding to the ECM, towards the nucleus. 

### 2.1. Impact of Microgravity on the Regulation of Genes Whose Products Act Extracellularly

Immunofluorescence staining (IFS) showed that the ECM protein fibronectin was redistributed in RPM-samples compared to control cells (Figure 3A,B). The *FN1* gene expression was not significantly altered (Figure 3C). In addition, fibronectin expression did not change in FTC-133 cells exposed to the RPM for 6 min (Figure 3D). *COL1A1* mRNA, however, was significantly upregulated in hyper-*g* samples compared with corresponding controls (Figure 3E), whereas the *COL4A5* mRNA remained stable in all groups (Figure 3F). The expression of the structural protein collagen IV was not significantly changed by short-term µ*g* as compared to the corresponding control (Figure 3G).

Moreover, we studied the ECM protein and cell adhesion molecule osteopontin. The secreted phosphoprotein 1 (*SPP1*) gene expression, determined in post-flight and hyper-*g* samples was slightly, but not significantly elevated (Figure 3H). RPM-exposure of the FTC-133 cells did not induce significant changes in the amount of protein in both groups (Figure 3I). qPCR of *TGFB1* mirrored these findings (Figure 3J).

Finally, we determined the gene expression patterns of tissue inhibitor of metalloproteinase 1 (*TIMP1*) and matrix metalloproteinases *MMP1*, *MMP3* and *MMP14* (Figure 3K-N). *TIMP1*, *MMP1*, and *MMP3* genes were slightly elevated in r-µ*g*, whereas the *MMP14* mRNA was elevated in r-µ*g* and significantly elevated in hyper-*g* samples (Figure 3N).

Earlier studies investigating the impact of µ*g* on human cells and the synthesis and secretion of the two cytokines interleukin-6 (IL-6) and interleukin-8 (IL-8 or synonymous CXC-motif-chemokine 8, (CXCL8)) have shown their involvement in 3D spheroid formation [24]. The release and expression of both cytokines was altered in space in FTC-133 cells [24]. Here we show that both, the *IL6* and *CXCL8* mRNA are elevated in the r-µ*g* and 18*g* centrifugation groups compared to controls (Figure 3O,P), while the IL8 protein expression was significantly enhanced during 6 min on the RPM (Figure 3Q). The *LOX* mRNA expression was elevated in r-µ*g* samples, but remained unaltered in all other groups (Figure 3R).

### 2.2. Influence of Microgravity on Genes of Membrane Proteins and Their Accumulation

The F-actin staining of RPM-exposed FTC-133 cells revealed stress fibers (yellow arrows) and lamellipodia together with filopodia (blue arrows) compared to controls (Figure 4A,B). In addition, F-actin accumulated at the cell membrane (Figure 4B). The beta-actin protein was reduced in RPM-samples compared to static 1*g*-controls (Figure 4C).

In a further step, we found a reduction for the protein ezrin in s-µ*g* (Figure 4D), but no significant changes for moesin (Figure 4E) [20].

Immunofluorescence staining of vinculin showed a normal distribution of this cytoskeletal protein, present in adherence-type cell junctions in the FTC-133 cells, at 1*g* (Figure 4F). A clear disorganization of vinculin was visible in RPM-exposed FTC-133 cells (Figure 4G). *VCL* was not changed in the flight samples, but significantly elevated by 18*g* centrifugation (Figure 4H). Vinculin expression was not changed after RPM-exposure compared to 1*g* (Figure 4I). Vinculin regulates the E-cadherin expression and is involved in mechanosensing by the E-cadherin complex. Therefore, we evaluated the *CDH1* mRNA expression, which was not altered in any of the samples (Figure 4J).

Vinculin and E-cadherin are known to be involved in cell morphology and locomotion. Another F-actin-binding protein is cofilin. The *CFL1* gene was elevated in r-µ*g*, and significantly enhanced in hyper-*g* samples (Figure 4K), indicating its role in the dynamics and organization of the F-actin cytoskeletal network. Western blot analysis of cofilin revealed a significant reduction in the RPM sample (Figure 4L). Focal adhesions are multiprotein structures connecting the ECM and the cytoskeleton.

Focal adhesion kinase 1 (FAK1 or protein tyrosine kinase 2 (PTK2)) is important for cell migration and its inhibition decreases mobility and metastasis. The *PTK2* mRNA was not altered in the TEXUS samples, but elevated by 18*g* centrifugation compared to the corresponding control (Figure 4M). The FAK1 protein content was significantly reduced in RPM samples (Figure 4N)

Finally, we measured the gene expression of the cell adhesion molecules vascular cell adhesion molecule 1 (*VCAM1*) (Figure 4O) and intercellular adhesion molecule 1 (*ICAM1*) (Figure 4R). Both proteins were upregulated in r-µ*g*, but not significantly changed in hyper-*g* (Figure 4O,R). ICAM-IFS visualized changes between the two groups (1*g* vs. RPM-exposure) (Figure 4P,Q). In RPM-samples ICAM1 IFS revealed holes and a lower density of ICAM-positive areas (Figure 4Q), but the ICAM1 protein concentration remained stable during 6 min on the RPM (Figure 4S).

Due to a kink in a fixative tubing one culture chamber was not filled with fixative, meaning that one µ*g*-chamber was not fixed with RNA*later* at the end of the 6-min µ*g*-phase [20]. Therefore, the cells harvested from the residual two μ*g*-samples did not allow a significant microarray gene analysis. In order to obtain preliminary hints on possible gene regulation within the six early minutes of µ*g*, we compared the result of microarray analysis of the three 1*g*-samples with the average values of two measurements of the µ*g*-samples. Of the 27,600 genes analyzed, eight genes appeared to be down-regulated, while seven seemed to be upregulated. While no further indication was found for the relevance of the apparent down-regulation, the comparison pointed to the possibility that *LOX*, *ADM*, *IGFBP3*, *NDRG1*, *KISS1R*, *GJB2*, *PFKFB4* genes could at least be upregulated1.7-fold after 6 min of µ*g*. The speculation is supported by the result that besides *PFKFB4* these genes and their products are members of the networks shown in Figure 1 and Figure 2. In addition, several earlier studies have shown that all these genes or their products are gravisensitive and are downregulated during a prolonged stay in weightlessness [11,24,25,26].

## 3. Discussion

The exposure of thyroid cells to r- and s-µ*g* led to promising results in regard to re-differentiation of TC cells [8,11,23,27,28,29]. Especially a 10-day long-term sojourn in µ*g* during an unmanned space mission revealed gene expression changes suggesting a re-differentiation of the TC cells compared to their ground controls [11]. In this study, we could advance this theory and learned that up- and downregulation of a given gene may be a time-dependent process.

### 3.1. Impact of Microgravity on the Cytoskeleton

It is still unknown how cells are able to sense µ*g*. An interesting and often discussed candidate for sensing and conduction of µ*g* is the cellular cytoskeleton [30,31,32,33]. For this reason, we focused here on important cytoskeletal factors, associated focal adhesion complex molecules, and actin-binding proteins, which are all interconnected at the protein level (Figure 2).

The actin cytoskeleton network is one of the most intensively investigated structures in µ*g*. It is involved in cell shape, migration and stress response, among others [34]. In previous studies we found that the β-actin expression in TC cells as well as in other cell types is remarkably regulated during short- and long-term exposure to s-µ*g* and r-µ*g* [8,11,23,27]. Following short-term r-µ*g* the *ACTB* gene expression was upregulated in a matter of seconds, for example [27]. These findings were emphasized by live-cell imaging during a sounding rocket flight, where F-actin cytoskeleton re-arrangements were monitored immediately after entering the µ*g*-phase [12]. Here, we investigated FTC-133 follicular TC cells exposed to a 6-min RPM-exposure and found similar changes which support the findings from the previous rocket mission TX52.

During the TEXUS mission, the *ACTB* gene expression was constant in all controls as well as after 1 min of hyper-*g,* either on the rocket or in the control-experiment [20]. However, the µ*g*-samples presented a distinct downregulation of *ACTB*, which is in line with findings from short-term experiments during a parabolic flight mission [12]. In contrast, after 7-day and 14-day-exposure of FTC-133 cells on the RPM, *ACTB* was found to be highly upregulated [8]. This indicates a strong regulatory effect on the cells as they start to form 3D cell aggregates [8].

Ezrin (EZR) and moesin (MSN) are part of a focal adhesion complex and interconnect the plasma membrane to the cytoskeleton [35]. Moesin forms a complex with ezrin and is able to bind to actin [36,37]. These proteins are strikingly involved in the surface structure adhesion, migration and have been associated with some human cancers [38,39]. Ezrin and moesin have also been found to be modulated in previous µ*g*-experiments with thyroid cells [12,28]. During a 24 h-exposure study on the RPM with the thyroid cancer cell line UCLA RO82-W-1, the *EZR* gene expression was significantly downregulated [28]. That is in alignment with the findings from the TEXUS flight, as only the expression in µ*g*-samples was downregulated while controls and hyper-*g* probes were not affected [20] and corresponds to the reduction of ezrin protein during 6 min on the RPM (Figure 4D). Interestingly, the *MSN* gene expression was slightly downregulated in µ*g*-samples and upregulated in hyper-*g* experiments. *MSN* was not found to be modulated during parabolic flights using microarray analyses [27]. However, a 24 h exposure on the RPM presented a downregulation in adherently growing cells, comparable to the present study [28].

### 3.2. Real Microgravity Changes the Extracellular Matrix

Furthermore, we investigated the ECM with a focus on fibronectin. Earlier detailed proteomic analyses have shown that FTC-133 cells express surface proteins that bind fibronectin, strengthening 3D growth [40]. These results are in agreement with our current data. We observed upregulated *FN1* mRNA in r-µ*g* samples. In a former study, ML1 follicular TC cells were exposed to parabolic flight conditions. It was demonstrated that the *COL4A5* mRNA was downregulated under r-µ*g*, whereas *OPN* and *FN1* were significantly upregulated after 31 parabolas [27]. After six min r-µ*g* the *SPP1* gene expression was clearly elevated in r-µ*g* samples (Figure 3H), but the *COL4A5* mRNA kept a stable expression during the TEXUS flight (Figure 3F), indicating that the duration of µ*g* exposure might play a role in the upregulation of this basement membrane gene. In addition, ML1 TC cells exposed to the RPM revealed an increased synthesis of collagen type I and III, fibronectin, laminin and chondroitin sulfate [41].

Lysyl oxidase (LOX; protein-lysine 6-oxidase), is a protein that is encoded by the *LOX* gene. LOX upregulation in cancer cells is of importance in tumorigenesis and may promote progression of the primary tumor. qPCR analysis displayed a *LOX* upregulation in the TEXUS r-µ*g* group compared to all other groups (Figure 3R), the findings were confirmed by microarray technology. The LOX enzyme supports the cross-linking of collagen and elastin, which are both ECM components [42] and *LOX* gene expression was identified as promoting culture flask adherence of EA.hy926 cells [25]. During the Sino/German Simbox/Shenzhou-8 spaceflight mission, where the ECM linker LOX was virtually switched off, very large multicellular spheroids were formed [43].

The collagen homeostasis is maintained via collagen synthesis and degradation by the TC cells and is influenced by matrix metalloproteinases (MMPs). The MMPs themselves are regulated by TIMPs [44]. Little is known about matrix metalloproteinases and follicular TC. TIMP-1 was identified as a biomarker candidate for papillary TC [45]. Recently, the TIMP-1 secretion of space-flown FTC-133 cells was measured as 2.8-fold elevated. In parallel, the MMP-3 release was 5.94-fold elevated in ISS space samples compared to 1*g*-samples [46]. These results contradict earlier results. Following the SimBox/Shenzhou-8 space mission we detected a reduced MMP-3 secretion in both, RPM-samples and in space-flown samples [11]. The ECM plays a key role in tissue maintenance and integrity [47]. MMPs degrade the ECM and are involved in cancer progression and MMP activity is regulated by TIMPs [48]. The mode of action of TIMPs in cancer is yet not known, but there are reports about tumor suppression and tumor progression properties [49]. Here we found an increase in *TIMP1*, *MMP1*, *MMP3* and *MMP14* mRNA expression in r-µ*g*-samples (Figure 3K–N), indicating an activation in orbit. These results support the data obtained from the ISS spaceflight in 2014 [46].

We detected an upregulation of the *TGFB1* gene expression in the r-µ*g* group as compared to 1*g*-control cells (Figure 3J). No change was found after centrifugation. TGF-β_1_ is a growth factor and a multifunctional cytokine. It is involved in biological processes like cell proliferation, adhesion and differentiation, and is therefore, a target in cancer therapy [50].

### 3.3. Alteration of Focal Adhesion Proteins

Focal adhesions or cell–matrix adhesions mediate cell signaling in response to ECM adhesion and serve as mechanical linkages to the ECM. Focal adhesions are known to be important gravisensors [51,52]. To determine alterations in focal adhesion proteins, we focused on vinculin as a first point of interest. Vinculin has a vital role in mechanotransduction with integrins at focal adhesion sites [53]. Vinculin also directly interacts with talin, integrins and actin, hence allowing a proper cellular migration and orchestrating focal adhesion [54]. In addition, vinculin is a component of the adherence junctions and mediates cellular and extracellular signals. The *VCL* gene expression was not altered by µ*g*, but elevated by centrifugation (Figure 4H). Structure and morphology of vinculin was visualized by IFS (Figure 4F,G). Interesting changes were visible by RPM-exposure (Figure 4G). Vinculin was disarranged in s-µ*g*, indicating a clear reaction to the low-gravity condition.

Another important focal adhesion factor is the cytosolic and mechanosensitive protein talin-1 (TLN1). It links integrin directly and indirectly via vinculin to the cytoskeleton. Integrins bind to talin and talin binds to vinculin and thus influence cell adhesion. In addition, the integrin receptors are involved in the attachment of adherent cells to the ECM. Talin is a mechanosensitive protein and is connecting integrins and the actin-cytoskeleton. It links actin to the integrin-β_1,_ which is inserted in the plasma membrane [55]. It is especially enriched in regions of cell-cell contacts and adhesion to substrate [56]. Because of its function, it is of high interest when cells transit from a two-dimensional (2D) cell growth to a 3D cell growth in µ*g*. These 3D-aggregates or multicellular spheroids resemble the in vivo situation much closer than conventional cell cultures, making them extremely valuable for cancer research [6,57]. A downregulation of *TLN1* in the early state of µ*g* might be an indication for detachment of the cells from the substrate [20]. On the background that talin has a vinculin-binding site and can recruit vinculin, it forms a complex with integrins and is thus inducing cell adhesion.

Integrin-β_1_ (*ITGB1*) is a subunit of a plasma membrane associate receptor family which primarily detects ECM components [58]. It is bound to and activated by talin [59]. *ITGB1* was not regulated in any experimental condition [20]. This is in in alignment with findings from short-term µ*g*-experiments conducted during parabolic flight missions as *ITGB1* was not regulated as well [27]. Interestingly, *ITGB1* was regulated when human Nthy-ori-3-1 cells were exposed to a RPM for 24 h suggesting that the regulation is cell type-, time- or µ*g*-origin-dependent [29].

In a next step, we focused on E-cadherin. Cadherins are Ca^2+^ -dependent cell adhesion molecules in the cell-cell adherence junction, which belongs to the plasma membrane and is connected to cytoskeletal actin filaments [60]. E-cadherin’s intracellular domains are connected to the cytoskeleton and its stability is regulated by catenin D1 [61], which can affect cell-cell adhesion [62]. E-cadherin is downregulated in tumors and multicellular MCF-7 breast cancer spheroids engineered by a 14-day RPM exposure and are known to be involved in metastasis [63]. The loss of E-cadherin expression in association with the epithelial–mesenchymal transition occurs frequently during tumor metastasis. The regulation of the adhesive activity of E-cadherin present at the cell surface by an inside-out signaling mechanism is important in cancer [64]. In this experiment, we found a slight increase in *CDH1* mRNA in the r-µ*g* group, but no changes in all other groups (Figure 4J). This is an interesting finding because the E-cadherin (*CDH1*) expression in cancer cells exposed to r-µ*g* is an unknown area. Proteomic pathway analyses demonstrated changes in papillary TC which were associated with the disruption of cell contacts (loss of E-cadherin), actin cytoskeleton dynamics and loss of differentiation markers, all hallmarks of an invasive phenotype [65].

We focused on the actin-binding protein cofilin, which disassembles actin filaments and influences the actin dynamics [66]. We determined the *CFL1* gene expression and found an elevation in the r-µ*g* group compared with the respective controls (Figure 4K). In addition, the gene was upregulated by 18*g*-centrifugation, while the protein concentration was reduced after 6 min on the RPM (Figure 4L). Little is known concerning the impact of cofilin in follicular TC. An interesting proteomic analysis was published recently [66]. The authors investigated the fine needle aspiration fluid protein patterns of papillary thyroid carcinomas. A statistically significant upregulation of cofilin-1 was detected among others [67]. Cofilin-1 is determining the direction of cell migration and has importance for metastasis in papillary thyroid cancer [68]. Future experiments will be necessary to study cofilin-1 in follicular thyroid cancer in detail.

Furthermore, we studied focal adhesion kinase 1 (protein tyrosine kinase 2, PTK2), which plays an important part during cell migration. Interestingly, *FAK1*/*PTK2* gene expression did not change by the sounding rocket flight, but it was upregulated by centrifugation. Recently, we showed that FTC-133 cells growing in monolayers or in spheroids after RPM-exposure incorporate vinculin, paxillin, focal adhesion kinase 1, and adenine diphosphate (ADP)-ribosylation factor 6 in different ways into the focal adhesion complex [69].

In summary, we were able show that focal adhesion proteins are important gravi-sensors and link the information to the ECM and the cytoskeleton.

In addition, we investigated the cell adhesion molecules ICAM-1 and VCAM-1. We had learned from the ISS CellBox-1 space mission that VCAM-1, TIMP1, protein kinase C_α_ and others are involved in the inhibition of spheroid formation under r-µ*g* [46]. Interestingly, both *ICAM1* and *VCAM1* mRNA expressions were slightly elevated in r-µ*g*-samples. They were not significantly altered after 18*g*-centrifugation or RPM-exposure. This finding might be due to the short µ*g*-exposure time.

### 3.4. Impact of altered gravity on cytokines

Centrifugation with 18*g* hyper-*g* and simulating the 31 parabolas of a parabolic flight induced the gene expression of *IL6* and *CXCL8* in ML1 TC cells [26]. Here, we found a similar result. Short-term 18*g*-centrifugation upregulated both *IL6* and *CXCL8* mRNAs. In addition, there was an increase detectable in both genes in the r-µ*g* group. These results confirmed earlier data, when TC cells were exposed to the RPM. We did observe that the *IL6* gene expression was enhanced in FTC-133 thyroid cancer cells, which remained adherent for 24 h on the RPM [23]. Similar results were obtained when human follicular epithelial thyroid Nthy-ori 3-1 cells were incubated for up to 72 h on the RPM [9]. Both cytokines have shown to have a 3D growth-promoting effect on follicular cancer spheroids [24]. In summary, the cytokines IL-6 and IL-8 are involved in migration and growth in thyroid cancer, and both are sensitive to altered gravity conditions. They seem to be involved in the initiation of MCS formation via focal adhesion proteins of benign cells [9] and cancer cells.

Taken together, our findings suggest that µ*g* is a much stronger trigger for gene expression changes than hyper-*g*. This enables us to adjust some controls in favor of µ*g*-samples in future space missions, which will result in increased scientific output and saves resources.

## 4. Materials and Methods

### 4.1. Cell Cultures

Cell culture and preparation of the experiments were performed as previously described in detail [20]. In short, FTC-133 poorly differentiated follicular thyroid cancer cells (Sigma Aldrich, St. Louis, MO, USA) were seeded into cell culture chambers at a defined density of 10^6^ cells/chamber one day prior to the rocket launch (Figure 5A). These cell culture chambers were placed into late access unites, which ensured introduction into the rocket very close to launch (Figure 5B). During flight the cells were exposed to a variety of stressors, besides an on-flight 1*g*-centrifuge (Figure 5C). To discriminate between effects of µ*g* and the different stressors the cell culture chambers were fled with RNAlater (Thermofisher Scientific, Waltham, MA, USA) fixative at defined time points (Figure 5D), resulting in samples to be analyzed for gene expression changes back in the home laboratory. Furthermore, FTC-133 cells were exposed to worst case-hyper-*g* on a centrifuge for 1 min which could have been the case during sounding rocket launch [20].

### 4.2. TEXUS Sounding Rocket Mission

Investigated conditions ranged from different 1*g*-controls, to hyper-*g*- and µ*g*-samples as described before [20]. In short, during a sounding rocket flight the launch phase of 1 min can reach an acceleration of up to 12*g* which could have an effect on the cells (sample: TX53 Hyper-*g*). After the launch, the payload enters a 6 min µ*g*-phase during which one part of the samples are kept on a 1*g*-centrifuge (sample: In-flight 1*g*-centrifuge), as a control, while the other part is exposed to µ*g* (sample: TX53 µ*g*). Shortly before re-entering the Earth atmosphere samples are fixed as described before. In addition to the flight samples, on ground 1*g* controls were performed with the cells kept horizontal (sample: TX53 Ground Control) and vertical (sample: TX53 In-flight 1*g* sim.). The vertical control was done due to the special arrangement of the cells on the in-flight centrifuge. Finally, a worst-case hyper-*g* experiment was performed keeping the cells on a centrifuge with 18*g* for 1 min (sample: Hyper-*g* 1 min).

### 4.3. Random Positioning Machine

Simulated µ*g* (s-µ*g*) was performed on the desktop RPM (Airbus Defence and Space (ADS), Leiden, The Netherlands) located in a standard incubator (37 °C and 5% CO_2_) as previously described [70].

In brief, the RPM was operated in real random mode with random interval and direction with a maximum speed of 12.5 revolutions per minute. In each case sample flasks were placed onto the middle frame with a maximum distance of 7 cm to the center of rotation providing a µ*g* quality in the range of 10^−4^–10^−2^
*g* (*n* = 5 samples each group/run) [10,71]. Samples were run on the RPM for 6 min to mimic the µ*g*-phase of the sounding rocket flight. The RPM-samples and the corresponding static 1*g*-control flasks were completely filled with medium. The 1*g*-samples were placed in the incubator next to the RPM (*n* = 5 samples each group/run).

### 4.4. F-actin Staining

F-actin was visualized by means of rhodamine-phalloidin staining (Molecular Probes^®^, Eugene, OR, USA). The nuclei were stained with Hoechst 33342 (Molecular Probes^®^). The method was published earlier [72,73].

### 4.5. Confocal Laser Scanning Microscopy (CLSM)

The stained samples were examined using a confocal laser scanning microscope (LSM 780, Zeiss, Jena, Germany) using 40× oil-immersion objective with a NA of 1.3 [74].

### 4.6. Immunofluorescence of Fibronectin, Vinculin and ICAM-1

Immunofluorescence staining was performed to visualize changes in fibronectin, vinculin and ICAM-1 proteins. After the experiments, the cells were washed three times with PBS and afterwards fixed in 4% PFA (Sigma-Aldrich) for 30 min at room temperature. The primary antibodies (fibronectin, (Invitrogen, Carlsbad, CA, USA), mouse, dilution 1:100, vinculin (Abcam, Cambridge, UK), mouse, dilution 1:200, ICAM1, (Cell Signaling Technology, Danvers, MA, USA), rabbit, dilution 1:200) were applied for 24 h. Then the slides were washed three times with PBS before incubation with secondary antibodies (AF488 conjugated anti-mouse/rabbit: 1:500, both Cell Signaling Technology) for 2 h. Nuclei were stained with DAPI (4′,6-diamidino-2-phenylindole) (Invitrogen) and mounted with Vectashield. Afterwards the slides were investigated by CLSM (LSM 780, Zeiss, Jena, Germany).

### 4.7. Western Blot Analysis

Western blotting was performed as described earlier [72,73]. In each experiment (performed three times) five different culture flasks were subjected to either s-µ*g* or 1*g*. At the end of the experiment cells were collected and solubilized in lysis buffer. Following lysis and centrifugation, aliquots of 30 µ*g* were subjected to SDS-PAGE and Western blotting. For the RPM-samples five lanes representing the AD cells and five lanes representing AD static 1*g*-control cells were loaded. The samples were loaded on Criterion XT 4–12% precast gels (BioRad, Hercules, CA, USA) and run for 1h at 150 volts. Proteins were then transfer with a TurboBlot (Biorad) (100 V, 30 min) to a PVDF membrane. Glycerine aldehyde-3-phosphate-dehydrogenase (GAPDH) was used as a loading control.

Membranes were then blocked 2 h in TBS-T containing 0.3% I-Block (Applied Biosystems, Foster City, CA, USA). For detection of the selected antigens (see Table 2) the membranes were incubated overnight at room temperature in TBS-T and 0.3% I-Block solutions of the antibodies. Following three washing steps of 5 min membranes were incubated additionally 2 h at room temperature secondary with a Horseradish peroxidase (HRP)-linked antibody (Cell Signaling Technology Inc., Danvers, MA, USA) diluted 1:4000 in TBS-T and 0.3% I-Block. The respective protein bands were visualized using BioRad Clarity Western ECL (BioRad) and images were captured with Image Quant LAS 4000 mini (GE Healthcare Life Science, Freiburg, Germany). Images of stained membranes were captured on Syngene PXi 4EZ image analysis system (Synoptics, Cambridge, UK) and analyzed using the ImageJ software (U.S. National Institutes of Health, Bethesda, MD, USA; http://rsb.info.nih.gov/ij/) for densitometric quantification of the respective bands and total protein load.

### 4.8. Postflight qPCR Analysis

qPCR analysis was performed as previously described [42]. In short, samples from the mission were transported and stored in RNA*later*. In addition, samples from hyper-g experiments were investigated. RNA was isolated from cells using the RNAeasy kit (Qiagen, Venlo, The Netherlands) following the manufactures instructions. After concentration determination via Nanodrop 2000, cDNA was produced using the first strand cDNA kit (Thermo Scientific, Waltham, MA, USA). qPCR was performed using the FAST SYBR^®^ Select Master Mix (Applied Biosystem) and the 7500 Fast Real-Time PCR System (Applied Biosystems) to determine the expression levels of the target genes (Table 3). The selected primers were designed to span exon-exon junctions and to have a Tm of 60 °C using NCBI Primer Blast and were synthesized by TIB Molbiol (Berlin, Germany). Samples were normalized to 18S rRNA and measured in triplicates. We used the comparative threshold cycle (ΔΔ*C*_T_) method for relative quantification of transcription levels, with “TX53 Ground Control” set as 100%.

### 4.9. In-silico Analyses

*In-silico* analyses were performed as described by Bauer et al. [22]. To investigate the mutual interactions of detected proteins, the UniProt accession numbers of the selected targets were induced in a Pathway Studio v.11 software (Elsevier Research Solutions, Amsterdam, The Netherlands) [63,69].

### 4.10. Microarray Analysis

Microarray analyses were performed as described by Kopp et al. [20].

### 4.11. Statistics

Statistical evaluation was performed using SPSS 15.0 (SPSS, Inc., Chicago, IL, USA). The Mann-Whitney-U-Test was used to compare the different conditions. All data is presented as mean ± standard deviation (SD) with a significance level of * *p* <  0.05.

## 5. Conclusions

Taken together, simulated hyper-*g* induced the gene expression of *COL1A1*, *VCL*, *CFL1*, *PTK2*, *IL6*, *CXCL8* and *MMP14*.

Unfortunately, after the TX53 mission, we noticed that only two r-µ*g* samples were fixed with RNAl*ater* [20], so that no statistical evaluation was possible. qPCR measured elevated mRNA of the ECM genes *FN1*, *SPP1*, *TGFB1*, *TIMP1*, *MMP1*, *MMP3*, *MMP14*. In addition, the cell adhesion genes *ICAM1* and *VCAM1*, the focal adhesion factors *CFL1* and *CDH1* as well as cytokines *IL6* and *CXCL8* were upregulated in r-µ*g* samples. All these factors have demonstrated their gravi-sensitivity. *FN1* and *TIMP1* are recently proposed as potential target genes in papillary TC [75]. Downregulation of E-cadherin plays a role in metastasis, and the restoration of E-cadherin inhibits tumor growth. Antibodies reducing E-cadherin activity favor MCS formation, and PP2, blocking the E-cadherin reducer SRC prevents MCS formation *in vitro* [63]. Therefore, E-cadherin seems to be an interesting target in follicular TC. Future detailed investigations focusing on E-cadherin/β-catenin signaling will be performed.

Furthermore, we were able to show that µ*g* achieved by a sounding rocket flight is a strong trigger for gene expression changes in FTC-133 cells. Future studies during TEXUS sounding rocket missions will be performed in order to increase the number of samples.

## Figures and Tables

**Figure 1 ijms-19-04001-f001:**
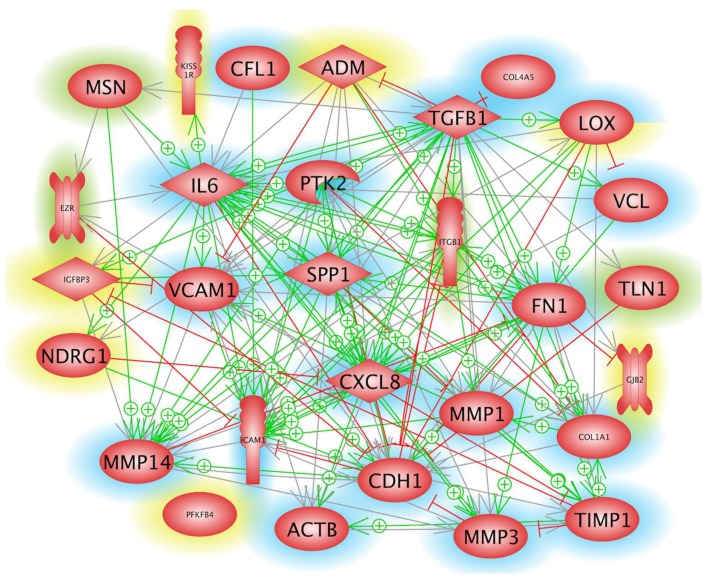
Network of interaction and mutual influence of the identified entities on a gene expression level. The genes, whose icons are marked with a green or blue rim have been found in recent studies referenced or mentioned in the preceding paper [20], respectively. The genes displayed with yellow icons attracted our attention when the two µ*g-*samples were investigated by gene array analysis, although the result lacked significance due to the fact of only two samples being available per experiments. Green arrows with a plus sign indicate expression or enhanced expression of a gene to which an arrow points to and red lines with a terminal crossbar show suppression or reduced expression of the gene near the crossbar. Grey arrows point to interaction with unknown effects as of now.

**Figure 2 ijms-19-04001-f002:**
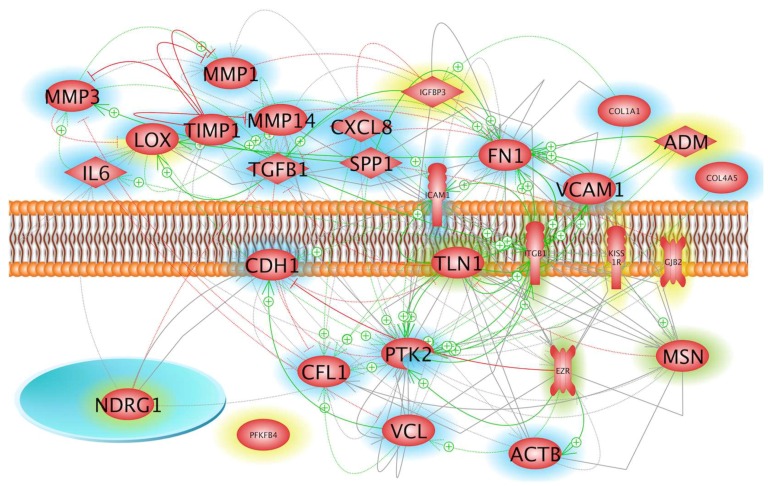
Localization and interaction of the products of genes mentioned above. The icons of genes products found in recent studies referenced or mentioned in the preceding paper [20], are marked with a green or blue rim, respectively. The genes products marked by yellow icons attracted our attention when the µ*g* samples (2) were evaluated by gene array analysis, although the results lacked significance due to the fact that only two samples per experiments were available. Green arrows with plus sign indicate stimulation and red lines with terminal crossbar show inhibition. Grey arrows indicate direct (solid line) or indirect (dashed line) interaction with unknown effect. Grey lines show protein–protein complex formation.

**Figure 3 ijms-19-04001-f003:**
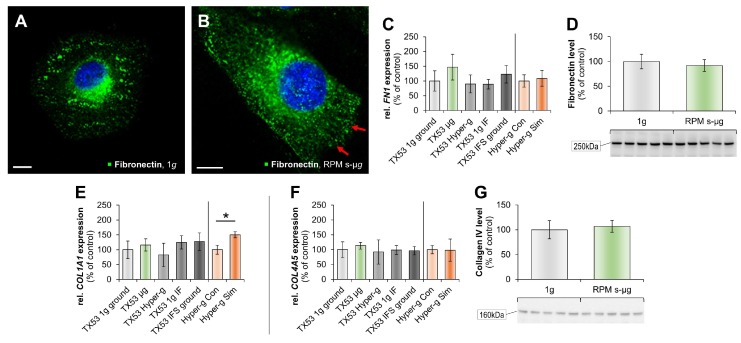
IFS of fibronectin (green) and nucleus (blue) (**A**,**B**); scale bar 10 µm; red arrow redistributed fibronectin, qPCR analyses of *FN1* (**C**), *Col1A1* (**E**), *Col4A5* (**F**), *SPP1* (**H**), *TGB1* (**J**), *TIMP1* (**K**), *MMP1* (**L**), *MMP3* (**M**), *MMP14* (**N**), *IL-6* (**O**), *CXCL8* (**P**) and *LOX* (**R**), and WB of fibronectin (**D**), collagen IV (**G**), osteopontin (**I**), and IL-8 (**Q**). “TX53” indicates samples collected from the sounding rocket mission, whereas “Hyper-g” presents data from a parallel experiment simulating the worst case-possible launch acceleration. IF: in-flight; Con: control; Sim: simulated; RPM: random positioning machine; 6min s-µ*g*; * *p* < 0.05 vs.1*g*.

**Figure 4 ijms-19-04001-f004:**
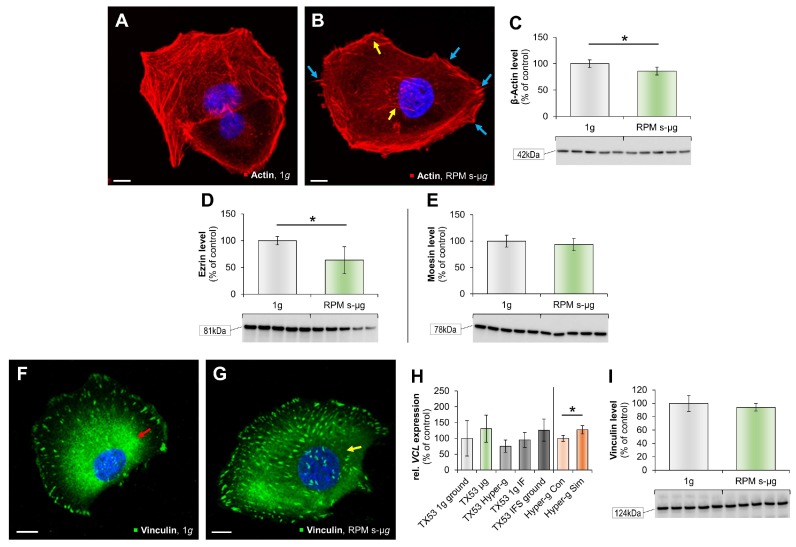
Histological staining of F-actin (red) and the nucleaus (blue) (**A**,**B**); yellow arrows: stress fibers and blue arrows: filopodia and lamellipodia; IFS of vinculin (green) and nucleus (blue) (**F**,**G**); red arrow: compact vinculin in 1*g* around the nucleus and yellow arrow: redistributed vinculin in RPM-samples; ICAM-1 (green) and nucleus (blue); yellow circles: areas with no ICAM1-positivity in RPM-samples (**P**,**Q**); scale bar 10 µm. qPCR analyses of *VCL* (**H**), *CDH1* (**J**), *CFL1* (**K**), *PTK2* (**M**), V*CAM1* (**O**), *ICAM1* (**R**), and WB of ACTB (**C**), ezrin (**D**), moesin (**E**), VCL (**I**), cofilin (**L**), FAK (**N**), ICAM1 (**S**). “TX53” indicates samples collected during the sounding rocket mission, whereas “Hyper-g” presents data from a parallel experiment aiming on the worst case-possible launch acceleration. IF: in-flight; Con: control; Sim: simulated; RPM: random positioning machine; 6min s-µ*g*; * *p* < 0.05 vs.1*g*, scale bars: 10 µm.

**Figure 5 ijms-19-04001-f005:**
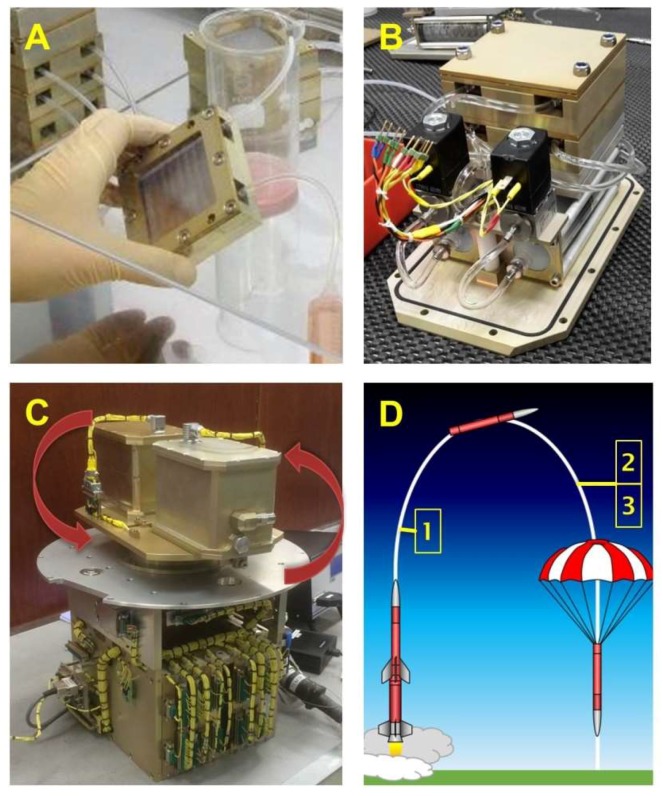
Workflow during the TEXUS-53 mission. Cell culture chamber were filled with cells (**A**) and afterwards placed into the late access unite (**B**). During flight a 1*g*-centrifuge (**C**) served as one control besides others. Red arrows indicate the rotation direction of the centrifuge. The cells were fixed at defined time points 1, 2, 3 (**D**).

**Table 1 ijms-19-04001-t001:** Summary of previously found regulations in selected molecular targets due to gravitational alterations. ↑ indicates upregulation; ↓ indicates downregulation; double arrow indicates significant alterations; → indicates no regulation, blank was not examined.

**Cell Line**	***g*-Condition**		***FN1***	***COL1A1***	***COL4A5***	***SPP1***	**Reference**
FTC-133	Space	10 days				↓	[11]
	RPM	1 day				AD↑↑ MCS↑	[23]
		7 days		↓		↑↑	[8]
		14 days		↓		↑↑	[8]
	PFC	31P				WT↑↑	[11]
	hyper-*g*	2 h					[11]
Nthy-ori 3-1	RPM	4 h	AD↑				[9]
		1 day	AD↑ MCS↓				[9]
		3 days	AD→ MCS→				[9]
		7 days		↑↑			[8]
		14 days		↑↑		↑↑	[8]
**Cell line**	***g*-Condition**		***ICAM1***	***VCAM1***	***IL6***	***CXCL8***	**Reference**
FTC-133	Space	10 days			↓	↓	[11]
	RPM	1 day			AD↑↑ MCS↓↓	AD↑↑ MCS↑	[23]
		7 days			↑↑	AD↑↑ MCS↓↓	[8]
		14 days			↑↑	AD↑↑ MCS→	[8]
	PFC	31 P			WT↑↑	WT↑↑	[11]
	hyper-g	2 h					[11]
Nthy-ori 3-1	RPM	4 h					[9]
		1 day					[9]
		3 days					[9]
		7 days			↑↑	AD↑↑ MCS↓↓	[8]
		14 days			↑↑	AD↑↑ MCS↑↑	[8]

**Table 2 ijms-19-04001-t002:** Antibodies for Western blot analyses.

Antibodies	Company/No	Species	MW (kDa)	Dilution
Collagen type I	Sigma/C2456	Ms	~130	1:500
B-actin	Sigma/A5316	Ms	42	1:2000
Fibronectin	Invitrogen/MA1198	Ms	250	1:1000
Osteopontin	Sigma/07264	Rb	50	1:1000
Cofilin	#ab 42824	Rb	20	1:2000
IL-8	Abcam/ab7747	Rb	11	1:500
Collagen type IV	Abcam/ab52235	Rb	160	1:1000
Ezrin	Cell Signaling #3145	Rb	81	1:500
Moesin	Cell Signaling #3150	Rb	78	1:500
Vinculin	Abcam/ab18058	Ms	124	1:1000
ICAM 1	Cell Signaling #4915S	Rb	89–92	1:500
GAPDH	Abcam/ab9384	Rb	37	1: 1000

**Table 3 ijms-19-04001-t003:** Primer sequences for qPCR.

Gene	Primer Name	Sequence
*18S-rRNA*	18S-F	GGAGCCTGCGGCTTAATTT
	18S-R	CAACTAAGAACGGCCATGCA
*ACTB*	ACTB-F	TGCCGACAGGATGCAGAAG
	ACTB-R	GCCGATCCACACGGAGTACT
*CDH1*	CDH1-F	GCTGGACCGAGAGAGTTTCC
	CDH1-R	CAGCTGTTGCTGTTGTGCTT
*CFL*	CFL-F	GAAGGAGGATCTGGTGTTTATCTTCT
	CFL-R	CCTTGGAGCTGGCATAAATCAT
*COL1A1*	COL1A1-F	ACGAAGACATCCCACCAATCAC
	COL1A1-R	CGTTGTCGCAGACGCAGAT
*COL4A5*	COL4A5-F	GGTACCTGTAACTACTATGCCAACTCCTA
	COL4A5-R	CGGCTAATTCGTGTCCTCAAG
*EZR*	EZR-F	GCAATCCAGCCAAATACAACTG
	EZR-R	CCACATAGTGGAGGCCAAAGTAC
*FAK1/* *PTK2*	FAK1-F	TGTGGGTAAACCAGATCCTGC
	FAK1-R	CTGAAGCTTGACACCCTCGT
*FN1*	FN1-F	AGATCTACCTGTACACCTTGAATGACA
	FN1-R	CATGATACCAGCAAGGAATTGG
*ICAM-1*	ICAM1-F	CGGCTGACGTGTGCAGTAAT
	ICAM1-R	CTTCTGAGACCTCTGGCTTCGT
*IL6*	IL6-F	CGGGAACGAAAGAGAAGCTCTA
	IL6-R	GAGCAGCCCCAGGGAGAA
*IL8/* *CXCL8*	IL8-F	TGGCAGCCTTCCTGATTTCT
	IL8-R	GGGTGGAAAGGTTTGGAGTATG
*ITGB1*	ITGB1-F	GAAAACAGCGCATATCTGGAAATT
	ITGB1-R	CAGCCAATCAGTGATCCACAA
*LOX*	LOX-F	TGGGAATGGCACAGTTGTCA
	LOX-R	AGCCACTCTCCTCTGGGTGTT
*MMP1*	MMP1-F	GTCAGGGGAGATCATCGGG
	MMP1-R	GAGCATCCCCTCCAATACCTG
*MMP3*	MMP3-F	ACAAAGGATACAACAGGGACCAA
	MMP3-R	TAGAGTGGGTACATCAAAGCTTCAGT
*MMP14*	MMP14-F	ACTTTATGGGGGTGAGTCAGG
	MMP14-R	GATGTTGGGCCCATAGGTGG
*MSN*	MSN-F	GAAATTTGTCATCAAGCCCATTG
	MSN-R	CCATGCACAAGGCCAAGAT
*PFN1*	PFN-F	GGGAATTTAGCATGGATCTTCGT
	PFN-R	ACCGTGGACACCTTCTTTGC
*SSP1*	SSP1-F	CGAGGTGATAGTGTGGTTTATGGA
	SSP1-R	CGTCTGTAGCATCAGGGTACTG
*TGFB1*	TGFB1-F	CACCCGCGTGCTAATGGT
	TGFB1-R	AGAGCAACACGGGTTCAGGTA
*TIMP1*	TIMP1-F	GCCATCGCCGCAGATC
	TIMP1-R	GCTATCAGCCACAGCAACAACA
*TLN1*	TLN1-F	GATGGCTATTACTCAGTACAGACAACTGA
	TLN1-R	CATAGTAGACTCCTCATCTCCTTCCA
*VCAM-1*	VCAM1--F	CATGGAATTCGAACCCAAACA
	VCAM1-R	GGCTGACCAAGACGGTTGTATC
*VCL*	VCL-F	GTCTCGGCTGCTCGTATCTT
	VCL-R	GTCCACCAGCCCTGTCATTT

All sequences are given in 5′–3′ direction.

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
