# Peer review of "Microgravity Affects Thyroid Cancer Cells during the TEXUS-53 Mission Stronger than Hypergravity"

_ijms, 2018, doi:10.3390/ijms19124001_

Round 1
Reviewer 1 Report
Major points
1. Line 107: ‘’Lower malignancy’’ should be described more concrete and scientifically.
2. Description of the Table 2 was not found in the text and it is needed.
Minor points
1. Line 49 and 59: ’’Random positioning machine’’ and ‘’tensegrity model’’ should be explained for general readers.
2. Line118: responds---response??, Line 122: life-cell imaging---lilve-cell imaging??
Author Response
We would like to thank the reviewers for their new input and for the opportunity to optimize our manuscript. We have tried to implement all requested changes and hope, that this version meets the reviewers’ standards.
Major points1. Line 107: ‘’Lower malignancy’’ should be described more concrete and scientifically.
Answer: The description was substantiated, the term lower malignancy was changed to redifferentiation (see lines 226 -230)
2. Description of the Table 2 was not found in the text and it is needed.
Answer: Table 2 was revised and changed in Table 1 and mentioned in the text (see line 94).
Minor points
1. Line 49 and 59: ’’Random positioning machine’’ and ‘’tensegrity model’’ should be explained for general readers.
Answer: We have performed investigations with the Random Positioning Machine in this manuscript to simulate 6 min microgravity. The term tensegrity was explained in lines 68-70. The method RPM is described in chapter 4.3, revised manuscript.
2. Line118: responds---response??, Line 122: life-cell imaging---lilve-cell imaging??
Answer: The errors were corrected in the revised manuscript.
Reviewer 2 Report
This study provides information on the effect of microgravity and hypergravity on the expression pattern of genes coding for cytoskeleton factors and associated focal adhesion complex molecules. The pattern has been studied in FTC-133 low differentiated follicular thyroid cancer cells. The authors confirm that microgravity, in contrast to hypergravity, affects the expression of the studied genes, and investigate by an in silico approach a possible interplay of the proteins coded by the studied genes.
Major points.
In a previous study of the TEXUS-53 mission the authors have reported on the expression of genes involved in several biological processes (Kopp et al., 2018, Scientific Reports), including the "not shown" data on gene coding for cytoskeleton factors and adhesion proteins, which are re-considered in this study. In this aspect, the reported data are not original. However, the authors performed an in silico analysis to identify possible mutual interactions among proteins coded by the investigated genes. This is an interesting point, which must be supported, in addition to the quantitative real time -PCR data, by the detection of the protein expression (western blot), and deserves both a punctual description of the results and a deep discussion.
In general, the design of manuscript does not lead to a prompt understanding of the aim and the
conclusions. The authors claim that "The purpose of this study was to find expression changes which might lead to the origin of malignancy". This implies that a comparison between malignant and non-malignant cells must be provided. This aspect is lacking, and, in fact, the conclusions seem to be totally disjointed from the declared aim. As it is evident that it is not possible to implement the research with data on nonmalignant cells, the authors might consider expressing the research aim differently.
In view of the conclusions, in which the authors stress a possible lowering of control samples in future studies (in contrast to the claimed "rigorous control experiments needed to be taken into account"), a detailed paragraph on the experimental design is required. By the way, in Table 1 the number of TX53 microgravity samples is not clear.
Table 2 is never cited along with the text, nor discussed. The authors should consider adding the results of the present study in Table 2, so that the readers can easily follow the discussion of the presented data in comparison with those previously published.
Minor points
Figure 1 should graphically report the statistically significant difference.
In Figure 2, some explanation is lacking (e.g. dashed and solid grey arrows are not described).
Author Response
We would like to thank the reviewers for their new input and for the opportunity to optimize our manuscript. We have tried to implement all requested changes and hope, that this version meets the reviewers’ standards.
This study provides information on the effect of microgravity and hypergravity on the expression pattern of genes coding for cytoskeleton factors and associated focal adhesion complex molecules. The pattern has been studied in FTC-133 low differentiated follicular thyroid cancer cells. The authors confirm that microgravity, in contrast to hypergravity, affects the expression of the studied genes, and investigate by an in silico approach a possible interplay of the proteins coded by the studied genes.
Major points.
In a previous study of the TEXUS-53 mission the authors have reported on the expression of genes involved in several biological processes (Kopp et al., 2018, Scientific Reports), including the "not shown" data on gene coding for cytoskeleton factors and adhesion proteins, which are re-considered in this study. In this aspect, the reported data are not original. However, the authors performed an in silico analysis to identify possible mutual interactions among proteins coded by the investigated genes. This is an interesting point, which must be supported, in addition to the quantitative real time -PCR data, by the detection of the protein expression (western blot), and deserves both a punctual description of the results and a deep discussion.
Answer: We have extended the study and performed a large number of new experiments. Further qPCR measurements, fluorescence stains and Western blots are included (Figs. 3, 4) and the in-silico analysis is extended (Figs. 1, 2). The description of the pictures was extended and the discussion was deepened and highlighted in yellow.
In general, the design of manuscript does not lead to a prompt understanding of the aim and the conclusions. The authors claim that "The purpose of this study was to find expression changes which might lead to the origin of malignancy". This implies that a comparison between malignant and non-malignant cells must be provided. This aspect is lacking, and, in fact, the conclusions seem to be totally disjointed from the declared aim. As it is evident that it is not possible to implement the research with data on nonmalignant cells, the authors might consider expressing the research aim differently.
Answer: The research aim was re-formulated and a large number of new experiments performed (see lines 28, 29). We have clearly described the aims of the study in the introduction, lines 76-80.
In view of the conclusions, in which the authors stress a possible lowering of control samples in future studies (in contrast to the claimed "rigorous control experiments needed to be taken into account"), a detailed paragraph on the experimental design is required. By the way, in Table 1 the number of TX53 microgravity samples is not clear.
Answer: This part was completely revised. The time points of taking samples and controls are illustrated in Figure 5 and described in lines 402 -412.
Table 2 is never cited along with the text, nor discussed. The authors should consider adding the results of the present study in Table 2, so that the readers can easily follow the discussion of the presented data in comparison with those previously published.
Answer: Table 2 was revised and changed in Table 1 and mentioned in the text (see line 93).
Minor points
Figure 1 should graphically report the statistically significant difference.
Answer: Fig. 1 was completely revised and partially included in the new Figs. 3 and 4. A large number of new results are given.
In Figure 2, some explanation is lacking (e.g. dashed and solid grey arrows are not described).
Answer: The lacking information was included in the legends of the new Figs. 1 and 2.
Round 2
Reviewer 2 Report
The authors significantly improved the manuscript that deserves to be published.